# RadBot-CXR: Classification of Four Clinical Finding Categories in Chest X-Ray Using Deep Learning

**Chen Brestel**
Zebra Medical Vision
chen@zebra-med.com

**Ran Shadmi**
Nucleai
ran@nucleaimd.com

**Itamar Tamir**
Rabin Medical Center & Tel Aviv University
dr.tamir.i.a@gmail.com

**Michal Cohen-Sfaty**
Zebra Medical Vision
michal@zebra-med.com

**Jonathan Laserson**
Zebra Medical Vision
jonil@zebra-med.com

**Eli Goz**
Zebra Medical Vision
eli@zebra-med.com

**Eldad Elnekave**
Zebra Medical Vision
eldad@zebra-med.com

## Abstract

The well-documented global shortage of radiologists is most acutely manifested in countries where the rapid rise of a middle class has created a new capacity to produce imaging studies at a rate which far exceeds the time required to train experts capable of interpreting such studies. The production to interpretation gap is seen clearly in the case of the most common of imaging studies: the chest x-ray, where technicians are increasingly called upon to not only acquire the image, but also to interpret it. The dearth of expert radiologists leads to both delayed and inaccurate diagnostic insights. The present study utilizes a robust radiology database, machine-learning technologies, and robust clinical validation to produce expert-level automatic interpretation of routine chest x-rays. Using a convolutional neural network (CNN) we achieve a performance which is slightly higher than radiologists in the detection of four common chest X-ray (CXR) findings which include focal lung opacities, diffuse lung opacity, cardiomegaly, and abnormal hilar prominence. The agreement of RadBot-CXR vs. radiologists is slightly higher (1-7%) than the agreement among a team of three expert radiologists.

**Keywords** CNN, Deep Learning, Medical Imaging, Algorithms, Computer Aided Diagnosis, CAD, Chest X-ray, CXR

## 1 Introduction

Chest X-rays (CXRs) are the most commonly performed of radiology examinations world-wide, with over 150 million obtained annually in the United States alone. CXRs are a cornerstone of acute triage, as well as longitudinal surveillance. Despite the ubiquity of the exam and its apparent technical simplicity, the chest x ray is widely regarded among radiologists as one of the most difficult to master[13].

Due to a shortage in supply of radiologists, radiographic technicians are increasingly called upon to provide preliminary interpretations, particularly in Europe and Africa. In the US, non-radiology physicians often provide preliminary or definitive readings of CXRs, decreasing the waiting interval at the non-trivial expense of diagnostic accuracy.

1st Conference on Medical Imaging with Deep Learning (MIDL 2018), Amsterdam, The Netherlands.

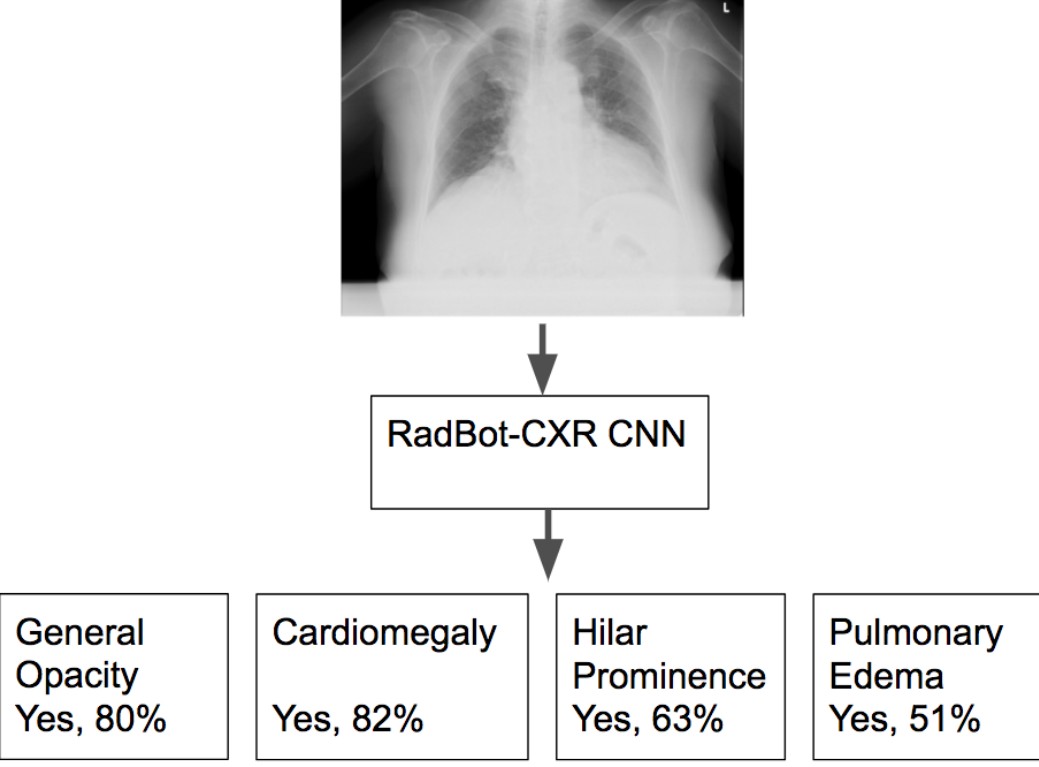

Figure 1: Input chest X-ray image is fed to RadBot-CXR CNN. The output reports the existence/non-existence for each of the findings: general opacity, cardiomegaly, hilar prominence and pulmonary edema.

Even among expert radiologists, clinically substantial errors are made in 3-6% of studies[13, 1], with minor errors seen in 30% [2]. Accurate diagnosis of some entities is particularly challenging: early lung cancer for example is missed in 19-54% of cases, with similar sensitivity figures described for pneumothorax and rib fracture detection. The likelihood for major diagnostic errors is directly correlated with both shift length and volume of examinations being read[4], a reminder that diagnostic accuracy varies substantially even at different times of the day for a given radiologist.

Therefore, there exists an immense unmet need and opportunity to provide immediate, consistent, and expert-level insight into every CXR. In the present work we describe a novel methodology employed for this endeavor and we present the results achieved using a robust method of clinical validation.

Clinically important CXR findings are commonly characterized by abnormalities within the lungs or the mediastinum; the four features selected in the present study reflect two subcategories of each. Within the lung, a subcategory of focal abnormality represents a broader spectrum of pathology including alveolar consolidation, atelectasis, pleural effusion and discrete lung mass. Pulmonary edema was chosen to represent diffuse lung abnormality. Within the mediastinum, cardiomegaly and abnormally prominent hilar opacities were selected for investigation.

## 1.1 RadBot-CXR

Our approach, RadBot-CXR (Figure 1), utilizes a CNN to detect the four imaging findings described above. Comparing the performance of RadBot-CXR to the a team of three expert radiologists shows that the agreement of RadBot-CXR vs. radiologists is slightly higher than the agreement among the radiologists team.

## 2 Previous Work

Automatic image analysis has been explored in recent decades utilizing computer vision and machine learning techniques. In 1998 Lecun et al. [8] demonstrated the use of LeNet 5 for character recognition with a seven layer network. Later in 2012, Krizhevsky et al. [7] introduced AlexNet, a convolutional neural network (CNN) which outperformed previous models in classifying natural images. Deeper networks were subsequently introduced, including Inception-V3 by Szegedy et al. [17]. Huang et al. incorporated features from prior layers in their DenseNet model [6].

CNNs have also been used for object detection and localization by Girshick et al.[3] and Ren et al. [12]. In addition, CNNs were later used for segmentation by Long et al. [10].

Medical image analysis has benefited from CNNs for classification, detection, segmentation and registration. A review on the field was done by Shen et al. [14]. Classification of a CT slice for predicting diseases was demonstrated by Shin et at. [15, 16]. Latent Dirichlet allocation (LDA) was used to generate disease labels for key slices based on textual reports. The disease labels are later used to train CNNs, both AlexNet and VGG 16 & 19, to predict a disease based on an input CT slice.

Classification and detection on chest X-ray images was demonstrated by Wang et al. [18]. Disease labels for eight clinical findings were generated by a natural language processing (NLP) approach. A CNN was then trained to classify and localize the clinical findings. Four different architectures of a CNN were tested. A recent work by Rajpurkar et at. [11] demonstrated the use of DenseNet for the classification of 14 clinical findings. For one of the findings, pneumonia, the performance of the model is also compared to radiologists. Another recent work by Li et al. [9] demonstrates the use of a ground truth with localization labels to predict both the existence of a clinical finding and also its location. The localization labels are supplied only for part of the data. Nevertheless, the work demonstrates an improvement in the classification performance as well.

## 3 RadBot-CXR

Our model is an inception-v3 variant CNN of 27 layers.

The input is a frontal (posterior-anterior, PA) CXR image, 8-12 bits, of a maximum size of 8.6 [MP] (3000*3000). Their original resolution is in the range (0.1,0.2) [mm/pix]. We first resized all images to 1024x1024 [pixels] and saved them in a local cache. We apply the following preprocessing. First, the raw pixel values are updated according to the *RescaleSlope, RescaleIntercept, PhotometricInterpretation* dicom tags. Next, the pixel values are stretched to the range (0,1). An algorithm to remove black frame from the image is then applied.

The input size of the network is 559*469 [pixels]. This is chosen to optimize the aspect ratio according to aspect ratio distribution in the DB. An interpolation is used to convert from size 1024*1024 [pixels] to 559*469 [pixels].

The output of the network includes multiple sigmoids. For each of the findings there exist one or three sigmoids. For unilateral findings such as cardiomegaly there is a single sigmoid. For bilateral findings, e.g. pulmonary edema, there are three sigmoids: one for the existence of the finding in the whole image, and additional two sigmoids for the existence of the finding on the left or the right sides.

## 4 Dataset

### 4.1 CXR Images

We utilized a dataset containing 1.5M frontal (PA) CXR studies obtained from adults over 18 years of age. The average age was 56.9 years. All studies were accompanied by an expert radiologist's interpretation.

Dataset generation ensued. We aimed to have several thousand positive and negative examples for each of the clinical findings. We employed manual textual searches to identify thousands of positive studies for each intended finding. Next, a random sample from the population was chosen to add a few thousand studies, of which the majority would be anticipated to be negative. Both the positive and negatives cases were matched for age and gender.

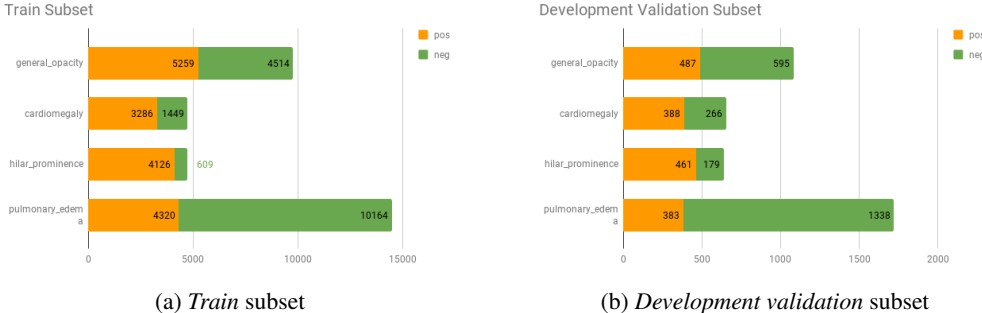

(a) *Train* subset           (b) *Development validation* subset

Figure 2: *Train* & *development validation* subset samples.

Each dataset was divided into training (80%), validation (10%) & test (10%) sub-sets. Each patient received an anonymized patient id used to associate the study with one of the subsets, to ensure that all studies of any specific patient reside in the same subset.

## 4.2  Image Tagging

A team of six radiologists analyzed the dataset images and tagged them using an in-house, web-based tagging application. On the tagging application, each image is shown together with a set of questions pertaining to the relevance (in terms of anatomy and PA orientation) of the image and to the presence or absence of the designated abnormal findings within the image. Where relevant (i.e., lung opacity and hilar prominence), laterality of the finding is also requested and tagged.

Each of the images in the training subset was tagged by a single radiologist, whereas, in the validation subset, each image was tagged by three radiologists. This was done to increase the confidence in the tagging by having multiple readers for each image. Each radiologists was blinded to the tagging of their peers.

7% of the images were tagged as non-relevant (an image is considered non-relevant if at least one of the radiologists tagged it as such).

The findings may have a partial correlation, i.e. an image may have more than one finding. Clinically, we know that there exists partial correlation between the findings. In addition, an image that serves as a positive example for one finding may serve also as a negative example for another finding.

Figure 2 summarizes the amount of samples we have for each of the findings. The *training* subset has 4.7K-14.5K samples for each finding. Among them 30-87% are positive samples, and 13-70% are negative samples. The *development validation* subset has 0.6K-1.7K samples for each finding. Among them 22-72% are positive samples, and 28-78% are negative samples. Another subset, *internal validation*, will be described in the next section.

## 5  Experiments

### 5.1  Training

The tagged samples are used to train a model using the following procedure. Beginning with a random selection of 32 images, random augmentation is applied on each image separately. Types of augmentations include: brightness, contrast, crop. Horizontal and vertical flips are excluded due to the asymmetric nature of the chest. Slight rotations were not tested.

The training procedure continues by iteratively choosing a random batch, and updating the weights by the rmsprops algorithm [5]. For every 1000 iterations, the performance of the model for each of the four findings was evaluated on the basis of several metrics.

One metric is inter-observer agreement among radiologists and between each radiologist and the algorithmic result. We have three tagging/votes for each image in the *development validation* subset.

Pair agreement was defined as follows. Given a set of tags by two taggers, their pair agreement is the fraction of cases the two taggers agreed. For example:

- 100 images were tagged
- 100 were tagged as relevant by both taggers
- 40 were tagged as positive by both taggers
- 50 were tagged as negative by both taggers
- Pair Agreement = (40 + 50)/100 = 0.90

We further define *cxr7_algo__rads* agreement as the average pair agreements of cxr7_algo and a radiologist. For example:

- cxr7_algo__rad1 agreement: 0.80
- cxr7_algo__rad2 agreement: 0.90
- cxr7_algo__rad3 agreement: 1.00
- Average: 0.90

In a similar manner we also define inter_rads agreement as the average pair agreements of radiologist *i* and radiologist *j*.

Another metric is sensitivity & specificity. We measure the point of equal sensitivity & specificity.

We conducted a few hundred training sessions. Various variants of architectures were tested.

Running on a Xeon E5-2650 v2 machine with 8 Tesla K80 gpu's, we had a training rate of 62 [images/sec]. A training session has 60k iterations, which result in 8.6 [h].

## 5.2 Choosing a Model & Model Calibration

Considering a few hundred training sessions with a few dozens models for each (measured every 1000 iterations), a few thousand putative models were generated. For each of the four findings, we chose an optimal model according to the metric of equal point sensitivity & specificity.

We then calibrated each of the chosen models separately for each findings. An expert radiologist used both an ROC curve and a scores bar display to calibrate the threshold for each finding. The scores bar display shows the images (with their scores) sorted by their scores.

## 5.3 RadBot-CXR Performance

The optimal four models are chosen using the sensitivity & specificity metric while measuring on the *development validation* subset. In order to measure the performance of the chosen models and verify their generalization, we conducted the following experiment.

We chose a random set of 20k images from the general population in our DB that did not belong to the training subset. We then ran the algorithm on this set. Inference running time on Xeon E5-2695 v4 machine is 330 [msec/image] when running on 4 GTX 1080Ti gpu's or 1 [sec/image] when running on a cpu. We then built an *internal validation* subset in the following manner.

*Internal validation*, Part I

- Randomly choose 126 images wherein the algorithm reports no findings
- For each finding, randomly choose 31 images wherein the algorithm reported a finding
- Total: 126 + 4*31 = 250 images

Part II is built in a similar manner, while excluding the images already chosen in Part I.

The tagging of the *internal validation* was done by three *expert* radiologists. Part I was used to measure the agreement when the radiologists do not know the algorithm result. In addition, Part II was used to measure the agreement when the radiologists do know the algorithm result.

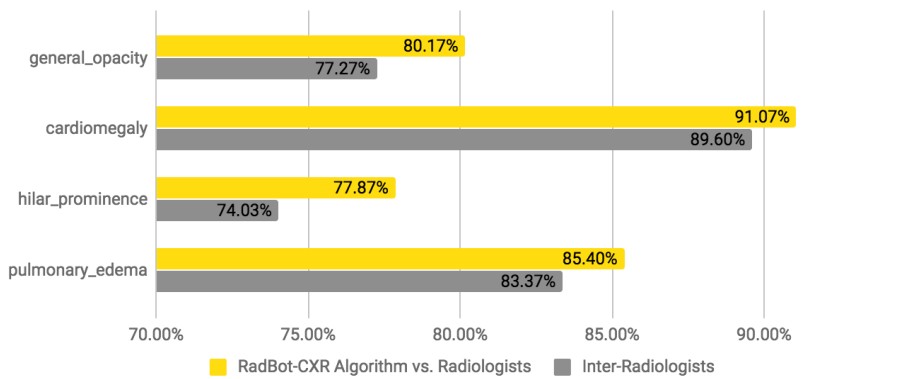

(a) Algorithm-radiologists agreement compared to the inter-radiologists agreement when the algorithm results are not shown. The agreement of the algorithm-radiologists is higher than the inter-radiologists by 1-4%.

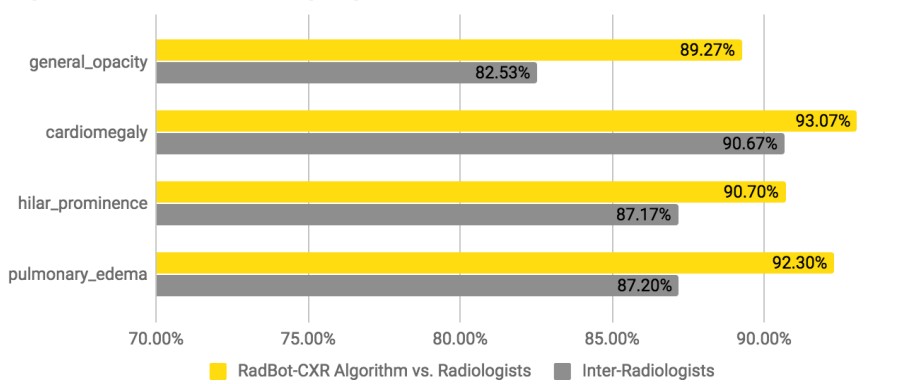

(b) Algorithm-radiologists agreement compared to the inter-radiologists agreement when the algorithm results are shown. The agreement of the algorithm-radiologists is higher than the inter-radiologists by 2-7%.

Figure 3: Agreement measurements.

Comparison of the algorithm-radiologists agreement compared to the inter-radiologist agreement is shown in Figure 3a for Part I. The inter-radiologist agreement was 74-90%, while the algorithm-radiologists agreement was 78-91%. The agreement of the algorithm-radiologists is higher than the inter-radiologists by 1-4%.

We further tested the algorithm performance on Part II while showing the radiologists the algorithm results. Comparison of the algorithm-radiologists agreement compared to the inter-radiologist agreement is shown in Figure 3b for Part II. The inter-radiologist agreement was 83-91%, while the algorithm-radiologists agreement was 89-93%. The agreement of the algorithm-radiologists is higher than the inter-radiologists by 2-7%.

We compared the performance measured on the two sets, Part I and Part II. When showing the algorithm results, both the algorithm-radiologists & the inter-radiologist agreements increased. The algorithm-radiologists agreement increased from 78-91% to 89-93%. Similarly, the inter-radiologist agreement increased from 74-90% to 83-91%. The inter-radiologist agreement increased by 1-13%, where hilar prominence agreement has the maximal increase by 13%. The algorithm-radiologists agreement increased by 2-13%, where again, the hilar prominence agreement had the maximal

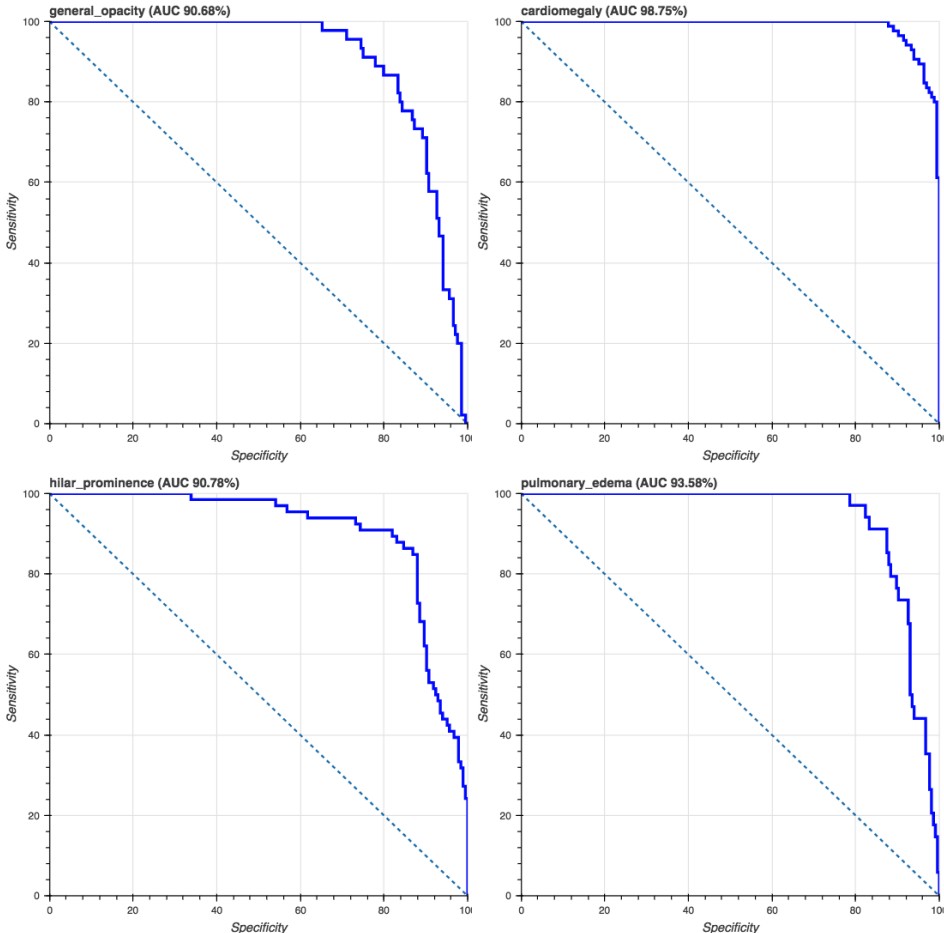

Figure 4: ROC measured on the *internal validation* for each of the meta-findings: general opacity, cardiomegaly, hilar prominence and pulmonary edema.

increase by 13%. General opacity and pulmonary edema agreements increased by 9% and 7%, respectively.

The agreement is a metric that is more appropriate than ROC to predict the value of such an algorithm in the field. Nevertheless, to compare ourselves to previous works that report AUC, we demonstrated the performance using this metric as well. Figure 4 shows the ROC graphs for the four finding categories, with AUC's 90.7-98.8%. As a ground truth label, we use the majority among the three radiologists' tagging.

We compared our AUCs to Li et al. [9] in Table 1. It is noted that our AUCs were measured on a different dataset since a preliminary assessment on a single finding showed a low agreement between our radiologist team and the NIH labels. Regarding cardiomegaly and pulmonary edema, our AUCs (98.8% and 93.6%) were higher by 11% and 5%, respectability. Li et al. do not detect hilar prominence, while our AUC is 90.8%. Finally, our general opacity AUC is 90.7%. Li et al. detect it as multiple findings with AUCs 79.5-86.7%. It is reasonable to assume their AUC for detecting general opacity meta-finding would be higher than 86.7%.

## 6 Conclusion

The present approach demonstrates a further advance in the challenging path to include automation in the medical care process. Using a CNN, we achieved a performance which demonstrate expert-level automatic interpretation of chest x -rays for seven distinct radiographic findings: alveolar consolidation, lung mass, atelectasis, pleural effusion; hilar prominence, diffuse pulmonary edema

Table 1: AUC's measurements with rough comparison to Li et al. [9], measured on different datasets. Regarding cardiomegaly and pulmonary edema, our AUCs are higher by 11% and 5% respectability

| finding | RadBot-CXR (ours) AUC [%] | Li et at. AUC [%] |
|---|---|---|
| Cardiomegaly | 98.8 | 87.4 |
| Pulmonary Edema | 93.6 | 88.2 |
| Hilar Prominence | 90.8 | — |
| General Opacity | 90.7 | ATL 79.6, CNS 79.5, EFF 86.7, MSS 83.1 |

and cardiomegaly. To the best of our knowledge, the present study is the first to assess inter-radiology concurrence and expert concurrence with algorithmic insights. The improved concurrence demonstrated after unblinding the expert to the algorithm decision indicates the potential for consistent algorithmic insights to improve diagnostic accuracy. The clinical implications of such performance are important, particularly in radiology-scarse settings, where a substantial portion of CXR's performed are not read by an expert radiologist.

Future and ongoing work will address localization of findings identified on the radiographic image, as well as expanding the number of findings detectable. Organ or anatomic compartement segmentation may further increase classification performance and/or accelerate training.

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
