# OpenReview forum: "RadBot-CXR: Classification of Four Clinical Finding Categories in Chest X-Ray Using Deep Learning"
_MIDL.amsterdam/2018/Conference — MIDL 2018 Poster_

### Review · AnonReviewer2 · 2018-04-27
**Classification of clinical findings in chest x-rays**

**Rating:** 1
**Confidence:** 2

**Review:**

This work explored the use of an Inception-v3 convolutional neural network variant for the classification of four clinical findings in chest x-rays. Although the results show good agreement with radiologists, it's unclear if the methods or results were validated robustly.

Some comments:
1) "A few dozen models were tested" - what models? were they all variants of Inception-v3?
2) What were your model parameters? What parameters did you experiment with?
3) What dataset did you use? Is it proprietary? Was it all acquired at the same institution? Dataset details should be included for fair comparisons with other datasets, as well as for reproducibility
4) It appears as though there are some severe class imbalances in the training set (particularly the hilar_prominence examples). This seems odd, since the training set was curated from the larger dataset.
5) Putting the results in a table would be beneficial for the paper, as its hard to pull out all of the information from in-text.
6) No standard deviation for results reported.
7) Results are compared with results from Li et al. on the NIH chest x-ray dataset, which is publicly available. Why not test on this as well, as it is also available to you? It isn't exactly fair to compare the two methods on two different datasets, and it doesn't make a strong claim that the proposed method is an improvement over Li et al.

Overall, the results seem promising, and I think the comparison of agreement  with experts is an excellent metric. For future work, more information on the model optimization, and more robust result reporting would be beneficial.

**Special Issue:**

No

---

### Review · AnonReviewer1 · 2018-04-30
**Review of RadBot-CXR**

**Rating:** 3
**Confidence:** 2

**Review:**

This paper presents an application of a CNN model (inception v3) on the classification of four types of clinical findings from Chest X-rays.  The main extension proposed is perhaps at the classification layer that contains two different types of classifications: binary classes (e.g., presence or absence of cardiomegaly), or a two-level classification with 3 nodes (e.g., presence or absence of pulmonary edema, and presence on the left or right sides - these additional classification nodes tend to improve classification, as already noted many times in the field).  The experiment was carefully implemented in several steps: 1) selection of positive and negative cases from a dataset containing 1.5M frontal (PA) CXR studies (patients over 18 years old) using a manual search of radiology reports; 2) this dataset was then analysed by a team of six radiologists, who tagged the images (into zero, one or more of the classes) using a web-based annotation system - training images were tagged by one radiologists, while validation/testing images were tagged by three radiologists; 3) the training of the CNN model is conducted for 60k iterations, and thousands of intermediate models are kept for evaluation; 4) models are then assessed and selected for each classification using the metric of equal point sensitivity & specificity on the validation set; and 5) the operating point for each classifier is defined manually by a radiologist using the ROC curve on the validation set.  Results show that the agreement of the proposed model with respect to the radiologists' majority vote on the test set tends to be higher than the agreement between radiologists.  Another experiment is done, where the radiologist can access the the result of the system before making his/her own classification, and higher agreement results are achieved (method vs radiologists and radiologists vs radiologists).  Finally a comparison in terms of AUC is done with the Li et al.'s, showing a better performance of the proposed method.

This is an interesting application paper, without any technical novelty, but with interesting results.  There are a couple of major issues, though.  First, the comparison with Li et al. is not fair because they used the (very noisy) annotation available from the NIH Chest X-ray dataset, while the annotation used by the proposed method looks a lot cleaner.  Also, why does the method rely on a radiologist to estimate the classifier's operating point?  This reduces the interest in the application as it is generally expected that the methodology can do that automatically.

Minor issues:
- The in-house, web-based application is similar to what is used in a clinical site?  If not this must be mentioned because it can affect the radiologist’s performance.
- What do the “Yes” and percentage mean in figure 1?  Pleas clarify.
- In the introduction, the paper mentions the proposal of a new methodology, but it does not specify the novelty.  Is it a new application?  A new technique?  New results? Please specify.
- Given that there may be errors in the radiologist’s interpretation of images, it’d be ideal for the experiments that an unequivocal ground truth was available.  A discussion around that topic should be present, together with a discussion on the fairness of the comparison with Li et al.'s method.
- Why does the conclusion refer to seven findings, while the whole paper focuses on four?  Please clarify.



**Special Issue:**

No

---

### Review · AnonReviewer3 · 2018-05-06
**New application of a known network on a very large dataset with some need to clarify the exact numbers used at different stages**

**Rating:** 4
**Confidence:** 2

**Review:**

The research work shows the applicability of inception-v3 based model trained to recognise positive and negative findings on chest X-rays.  The models are trained on a data set, which was generated form 1.5M studies ensuring a match for gender and age of patients. The main issue was the lack of clarity with respect to the number of patients used at different stages of the approach.

The models were validated for the agreement with radiologist opinion in twofold: with and without showing models results to radiologist during the decision-making process. Authors showed improvement in regards to radiologist performance and comparison with other method presented in 9.

Overall, the authors tackle an interesting topic in decision support systems. They build a model which might play a role in the diagnostic process. They showed improvement in the decision-making process. They do not however discuss a problem of cognitive bias such as confirmation bias introduced by having the results of the algorithm presented first to the human rater.  They successfully compare they result to another work citation 9. showing that RadBot-CXR outperforms the method proposed in 9.

Clarity: The main idea of this research is fairly well stated.  There are some typos and grammatical errors. The main issue relates to the clarity on the dataset. Given how important the dataset is for this work, the authors should make all the data descriptions crystal clear.

Also the experiments could be presented better.
.    The authors use  inception-v3, but they do not provide all necessary hyperparameters used for training, e.g. learning rate, the momentum of rmsprops optimizer, loss function. The authors stated “Various variants of architectures were tested”, but they do not explain what variants exactly. It needs more explanation.
.    The authors use non-specific descriptors for their methods such as in section 5.2 “Considering a few hundred training sessions with a few dozens models for each (measured every 1000 iterations), a few thousand putative models were generated.”
They should provide specific numbers and explain clearly how they choose the best model.
.    They also do not provide details for “An algorithm to remove black frame from the image is then applied.” and  “An interpolation is used to convert from size 1024*1024 [pixels] to 559*469 [pixels]”. More details would make their work reproducible.
.    They do not strictly define “the point of equal sensitivity & specificity.”
.    They train models for 60k iteration on 20k images. They do not provide any details which confirm model convergence e.g. training loss. They also state that they check models every 1000 iteration and choose the most optimal model based on sensitivity and specificity.  It is not clear from that discerption how long final model was trained.
.    They defined pair agreement between radiologist as a sum of the same indication for images divided by the number of images. They do not consider the chance agreement. I would consider more robust statistic e.g. Kappa statistic, correlation coefficient.
.    They train models for every clinical finding separately having positive and negative clinical examples. In figure 2 they show a number of samples in the class used for the training. The ratio of positive to negative finding is very biased towards one of the class, eg. Pulnamary_edama 4320 vs 10164 samples for positive and negative cases, respectively. They do not provide any details how they deal with the training of the network in regards to not equal class size , which can contribute to a biased model for bigger in size class. It would be interesting to report on the use of weighted cross-entropy as a loss function.


Methodological problem.
The authors do not sufficiently discuss the confirmation bias introduced by having the result of the algorithm used as an observed first rater by the radiologists. They show that radiologists agree more with each other and with the algorithm when the algorithm results are known a priori to them. It is rather expected that higher agreement score is achieved due to nature of decision-making process under confirmation bias. This should be better discussed especially in light of the recognised usefulness of ROC curves which is somewhat dismissed in this work..

The authors also stated that “ It is noted that our AUCs were measured on a different dataset since a preliminary assessment on a single finding showed a low agreement between our radiologist team and the NIH labels”. This might indicated a lack of generalisability to the external dataset, which impact should be discussed in light of the targeted clinical use.


**Special Issue:**

Yes

---

### Decision · Program_Chairs · 2018-05-15
**Paper1 Acceptance Decision**

Poster